# Increasing the Corrosion Resistance in the UNS S32750 Super Duplex Steel Welded Joints through Hybrid GTAW-Laser Welding and Nitrogen

**DOI:** 10.3390/ma16020543

**Published:** 2023-01-05

**Authors:** Arthur M. Videira, Willians R. Mendes, Vicente A. Ventrella, Irene Calliari

**Affiliations:** 1Department of Control and Automation Engineering, Federal Institute of Education, Science and Technology of Mato Grosso, Primavera do Leste 78850-000, Brazil; 2Department of Mechanical Engineering, São Paulo State University, Ilha Solteira 15385-000, Brazil; 3Department of Industrial Engineering, University of Padova, 35131 Padova, Italy

**Keywords:** super duplex, pulsed laser, GTAW, hybrid welding, UNS S32750, austenite

## Abstract

The development of techniques to improve the welding of super duplex steels is necessary in order to ensure that the phase balance and properties of the material are not affected during this process. Hybrid arc-laser welding is a perfect combination of the advantages of both processes, producing deeper weld beads with more balanced phases than the pulsed laser process. Here, the objective was to improve the corrosion resistance of UNS S32750 weld beads by increasing the volumetric austenite percentage in the fusion zone (FZ) with a hybrid process of GTAW (gas tungsten arc welding) and pulsed laser Nd-YAG (neodymium-doped yttrium aluminum garnet). Welds were performed in bead on plate conditions with fixed laser parameters and a varying heat input introduced through the GTAW process. Additionally, welds within a nitrogen atmosphere were performed. After base metal characterization, an analysis of the FZ and heat affected zone were performed with optical microscopy, scanning electron microscopy and critical pitting tests (CPT). The synergy between the thermal input provided by the hybrid process and austenite-promoting characteristic of nitrogen led to a balanced volumetric austenite/ferrite fraction. Consequently, the results obtained in CPT tests were better than conventional welding processes, such as laser or GTAW solely.

## 1. Introduction

Super duplex stainless steels have characteristics that combine good mechanical properties and corrosion resistance, as they consist of a two-phase structure with 50% ferrite (δ) and 50% austenite (γ) volumetric fractions. Therefore, the development of techniques to maintain the properties of super duplex steels is necessary to ensure that the phase balance and material properties are not severely impaired in this process.

The welding process leads to microstructural changes, unbalancing the phase fractions in the welded joint and surrounding areas, reducing the mechanical and corrosion resistance in these regions. Moreover, cracks may occur due to the thermal cycles involved in the welding process, with localized heating and cooling, and expansion and contraction [1]. Welding processes, such as GTAW (gas tungsten arc welding) and pulsed Nd-YAG Laser (neodymium-doped yttrium aluminum garnet), can be applied in super duplex plates, as long as welding parameters that promote phase balance are used. Previous studies showed that the use of these processes, controlling only the heat input, produce microstructures with a higher percentage of ferrite [2,3]. The reason is the very high cooling rate, avoiding the transformation of primary ferrite into austenite [4,5]. Increasing the amount of δ in the welded joint is detrimental to the corrosion resistance characteristics for several reasons.

Ferrite has lower pitting resistance equivalent numbers (PREN) than austenite, as it has lower levels of N, favoring the pitting corrosion and also promoting lower values in critical pitting temperature (CPT) tests [6,7,8].

The possible existence of depleted Cr regions due to the precipitation of Cr nitrides close to the δ grain boundaries is critical for corrosion properties as well. Furthermore, δ is rich in ferritizing elements (Cr, Mo, Si) and its formation during the welding cycle can lead to a decrease in the Cr content in adjacent metal phases, which justifies the lower corrosion resistance when there is more ferrite than austenite in the fusion zone (FZ) [6]. Another reason for ferrite to be controlled at the welded joint is that it reduces toughness at low temperatures [7].

As shown in the Cr–Ni–68%Fe pseudobinary diagram (Figure 1), the solidification of super duplex steels starts with primary ferrite, which turns into austenite [3,7,8,9]. The addition of nitrogen in the molten pool favors the formation of austenite because it acts by displacing the solvus line of the Cr–Ni–68%Fe pseudobinary diagram [10], increasing (hatched region) the biphasic area (austenite (γ) + ferrite (δ)) in Figure 1.

In other words, the addition of nitrogen in the FZ of the material provides more time for the ferrite (δ) to become austenite (γ), increasing the volumetric percentage of γ present in the weld bead.

A previous study [1] constructed the temperature profile during laser welding on a duplex steel sheet, concluding that the material cools down quickly, inhibiting the formation of austenite in the welded joint, as the FZ reaches 300 °C in less than 1 s after welding, being far from the austenite formation temperatures shown in Figure 1. In this sense, the graphs in Figure 2 demonstrate the effects of nitrogen in increasing the time for γ formation during Nd-YAG pulsed laser welding, hybrid GTAW–pulsed laser and hybrid GTAW–pulsed laser with nitrogen addition. Furthermore, the increase in heat input ensures a greater absorption of nitrogen in the FZ [12].

When comparing the graphs in Figure 2, we conclude that adding nitrogen to the hybrid GTAW pulsed laser process can be an excellent solution: with the heat input from the GTAW process, which already promotes the formation of austenite, there will be a longer time and more energy for nitrogen absorption, promoting a higher percentage of austenite in the FZ, improving its resistance to corrosion and, at the same time, obtaining a deeper penetration with the pulsed laser process.

Then, the objective was to obtain, through the hybrid GTAW (gas tungsten arc welding)/pulsed Nd-YAG (neodymium-doped yttrium aluminum garnet) laser welding process, a volumetric fraction of at least 50% austenite in the FZ and, consequently, better results in critical pitting temperature (CPT) tests than the single pulsed laser welding in the super duplex stainless steel UNS S32750.

The novelty of this work is to demonstrate the viability of hybrid GTAW laser welding to control the volumetric fraction of the ferrite/austenite phases in the FZ, revealing that it is possible to obtain balanced phases in super duplex steel sheets with a continuous welding process.

## 2. Materials and Methods

The base material was a UNS S32750 super duplex stainless steel sheet of 3.0 mm thickness with nominal chemical composition shown in Table 1.

The weld beads were made with a Nd-YAG pulsed laser (neodymium-doped yttrium aluminum garnet) process (LA and LN), and a hybrid GTAW (gas tungsten arc welding)/Nd-YAG pulsed laser (H20, H40, H20N and H40N) process, all in bead on plate conditions (“N” indicates samples with nitrogen addition). To perform the hybrid welding, a GTAW torch from a rectifier source was assembled with the laser turret and N_2_ nozzle. They worked simultaneously according to the parameters set out in Table 2. The process was automated through a holder that fixed the plate to be welded, moving it under the GTAW torch, N_2_ nozzle and laser aim in a linear movement at a constant and controlled speed, as shown in Figure 3a. The laser aim was focused on the surface and directed immediately after the GTAW torch, so that it shot after formation and into the GTAW molten pool in the metal, providing greater and deeper heating of the sheet and increasing N_2_ absorption. The working distance was 2 mm (Figure 3b).

The required conditions for the pulsed laser process were defined as the parameters that obtained an approximate depth of 1.5 mm (half the thickness of the UNS S32750 steel plate). In this way, it was possible to reach a minimum penetration for the sheet to be joined on both sides through the proposed process. These parameters were reached experimentally, controlling the frequency, temporal width and peak power, so that weld bead depths of 1.5 mm in the plate were obtained at a rate of 70% overlap. Furthermore, previous works were considered to establish the parameters of the welding processes [13,14,15,16,17]. The GTAW setup was an AWS EWTh2 tungsten electrode with 2.0 mm diameter, 45° angle, 2 mm arc length and direct current. While welding, the voltage was around 10 V and the heat input from GTAW process was considered with a 75% efficiency [18]. Table 2 shows the welding conditions for all samples.

Then, each weld bead was cut into three sections, which were metallographically prepared using sanding and polishing, providing three samples of each welding condition. The analysis of the base metal, fusion zone and heat affected zone (HAZ) was performed using optical microscopy (OM) and scanning electron microscopy (SEM), after etching these samples with Beraha’s reagent for 10 s [19]. The phase balance in volumetric fraction percentage (%V) was evaluated with the software ImageJ, with the average and standard deviation of the three sections of each condition obtained. In the same way, their geometry was measured and calculated. In sequence, the top surface of each weld bead was cleaned and protected with a heat resistant varnish, leaving a delimited area of 1 cm² exposed for critical pitting temperature (CPT) tests. Finally, the CPT tests were performed according to ASTM G150 standards [20] on the surfaces of each weld bead.

## 3. Results and Discussion

### 3.1. Base Metal

The characterization of the base metal as received was carried out by analyzing the volumetric fraction of ferrite (δ) and austenite (γ). Observations using MO and SEM revealed a volumetric fraction of each phase of around 50% and a typical microstructure of super duplex stainless steel (Table 3), consisting of a ferrite matrix with elongated austenite islands in the lamination direction (Figure 4), as obtained by previous authors [21,22]. The CPT test (Figure 5a) obtained a value of 92 °C (Figure 5b), which was compatible with the literature [23,24,25]. Thus, the base metal was in perfect condition for the proposed study.

### 3.2. Weld Beads

All weld beads presented a good surface finish without visual defects, but the weld beads with nitrogen addition (LN, H20N and H40N) had a lighter and brighter surface (Figure 6). In Table 4, there is a summary of the results obtained, while Figure 7 shows the cross sections of each weld bead.

Comparing the weld beads geometries, the hybrid process was able to produce joints with a higher depth when compared to the single laser process, allowing the welding of thicker plates (Table 4). The microstructure in the FZ of weld beads LA and LN was composed of primary ferrite (δ), whose orientation followed the cooling direction during welding (Figure 7). These characteristics were in accordance with the descriptions of previous authors [26,27], who claimed that in materials with a high Cr/Ni ratio the solidification in FZ firstly produces only ferrite, followed by nucleation of austenite at the grain boundaries of the already solidified ferrite, in a solid-state transformation (Figure 8a).

The pulsed laser conditions for super duplex steels are known for producing a weld bead with a low content of austenite [2] and a FZ microstructure based on a ferritic matrix and much less austenite than the base metal. In addition, Figure 8b (LA) shows intragranular austenite bands due to overlapping laser pulses, which provided more energy for their nucleation inside the grains [28]. The heat input and the short time of the laser pulse did not keep the material at the required temperature for long enough for the large scale transformation of ferrite to austenite. Some regions of the FZ did not even show visible austenite grains in OM (Figure 9a). Comparing LA and LN conditions (Table 4), there was no significant difference in the austenite %V results, because in the LN condition, the short laser pulse time and the low heat input were not sufficient to allow the N_2_ to interact with the welding pool. Thus, in all the weld beads formed with pulsed laser only, the %V of austenite, considering also specific regions of the FZ, varied from 0.01 to 8%, which corroborated findings from previous studies that showed a decrease in austenite %V in the FZ in processes with low heat inputs and fast cooling rates [13,14,15,16,17].

In Figure 9b the HAZ was very small, being imperceptible, and it was not possible to delimit it. Therefore, we confirmed that the pulsed Nd-YAG laser welding on UNS S32750 steel produced unbalanced joints. The high volumetric fraction of ferrite found in the FZ (Table 4 and Figure 7) can be detrimental to corrosion resistance and may have caused the poor performance in CPT tests, where a lower content of austenite in the FZ resulted in lower CPT values of around 26 °C for LA and 27 °C for LN.

In another way, when applying the hybrid process in the H20 and H40 conditions, the heat input from GTAW provided more energy to the solid-state transformation of ferrite to austenite, increasing the austenite %V in the FZ, as shown in Table 4 and Figure 10 (H20 and H40). In the H20N and H40N conditions, the N_2_ flow directed to the metal molten pool resulted in a higher %V in the FZ (Figure 10) because the GTAW welding pool provided more time and a higher energy for the nitrogen to penetrate in the FZ [12], while the laser shots allowed the nitrogen to penetrate deeper into the weld bead, as shown in Figure 7, where each laser pulse in the H20N condition left one deeper austenite layer in the section. This phenomenon is clear to see when comparing the H20N and H40N conditions: increasing the GTAW current in the weld bead (Figure 10) produced a high content of austenite, reaching around 50% of each phase.

As in the LA and LN conditions, the microstructure morphology in the FZ of hybrid weld beads showed epitaxial growth of the grains in the cooling direction (Figure 7, Figure 8 and Figure 9). The morphology of austenite present in the hybrid weld’s FZ (Figure 10) was also characteristic of intergranular and intragranular nucleation, but it also had some regions with austenite growth occurring in plates, known as Widmanstatten γ. The latter does not form at lower heat inputs, such as those found in LA and LN weld beads, because it needs more energy, as it results from the detachment of austenite plates previously formed in the grain boundaries (intergranular γ) [13,28,29,30].

The results of the CPT tests were improved with the hybrid welding (Figure 11) due to the increase in the %V of austenite in the FZ, reaching 37 °C (H20) and 45 °C (H40). With the addition of N_2_ in hybrid welding, values of 67 °C (H20N) and 74 °C (H40N) were obtained for the same reason. Thus, N_2_ and an increased thermal input promoted better results in the CPT tests.

In brief, when compared to the pulsed laser or GTAW processes [2,31,32], all hybrid welds with nitrogen addition presented a higher volumetric fraction of austenite in the FZ, resulting in a good corrosion resistance, reaching higher values (74 °C in the H40N) than the pulsed laser welding (Figure 11). The hybrid welding with N_2_ addition proved to be effective to promote the interaction of the N_2_ with the fusion pool, increasing the austenite %V in the fusion zone, reaching more than 50%, and improving the corrosion resistance of the joint.

## 4. Conclusions

The low heat input of pulsed laser welding did not allow primary ferrite to transform into austenite. Furthermore, the energy was not sufficient for nitrogen to interact with the welding pool, meaning that its addition was not effective. Consequently, the pulsed laser conditions analyzed produced weld beads with a low content of austenite, impairing the corrosion resistance and achieving low values in the CPT tests.

In comparison to the pulsed laser welding process with the same parameters, the greater heat input of the GTAW process, which keeps the metal temperature in the range of austenite formation for longer, resulted in a higher volumetric percentage of austenite when applying the GTAW pulsed laser welding in UNS S32750. Furthermore, the hybrid joints corrosion resistance was improved through hybrid welding, as shown in the CPT test results. The hybrid process allowed the nitrogen to interact with the welding pool and proved that nitrogen addition can be beneficial to the fusion zone, increasing the austenite volumetric fraction and corrosion resistance, and achieving a better performance in the CPT tests. Moreover, the hybrid welding promoted deeper fusion zone penetration, producing deeper weld beads when compared to using GTAW or pulsed laser only. The morphology of austenite present in the FZ of all weld beads was characteristic of intergranular and intragranular nucleation, but in the hybrid welding the presence of Widmanstatten γ was noticed due the higher heat input.

The results showed an innovative manufacturing process with known parameters providing excellent corrosion properties in the welded beads. The correct adjustment of welding conditions can promote a high production rate in industry, as the welded joints could be produced with the desired properties without the need for subsequent heat treatments.

For future works, we suggest obtaining the elemental composition of the FZ. This could be useful to verify the amount of nitrogen absorbed. Moreover, performing hybrid welding beads with an increase in the heat input above that used in this work would allow thicker plates to be welded.

## Figures and Tables

**Figure 1 materials-16-00543-f001:**
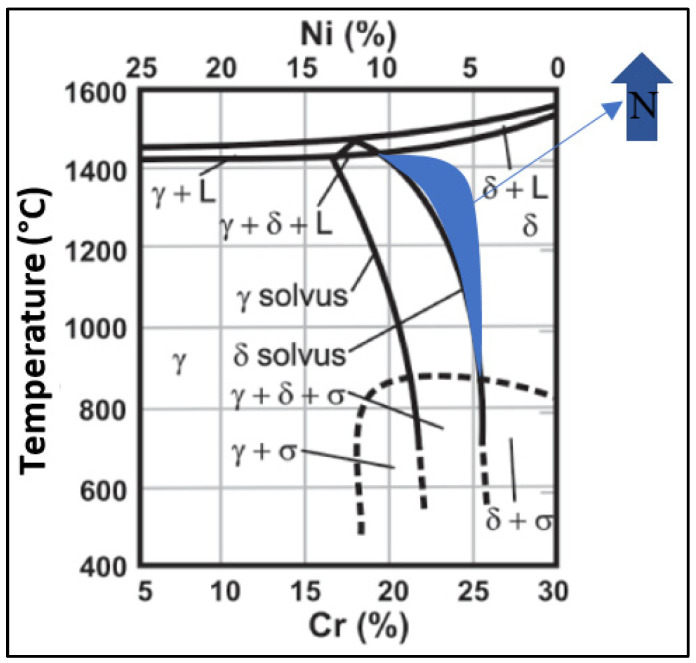
Cr–Ni–68%Fe pseudobinary diagram adapted by the authors [11]. The addition of N_2_ expands the biphasic δ + γ field.

**Figure 2 materials-16-00543-f002:**
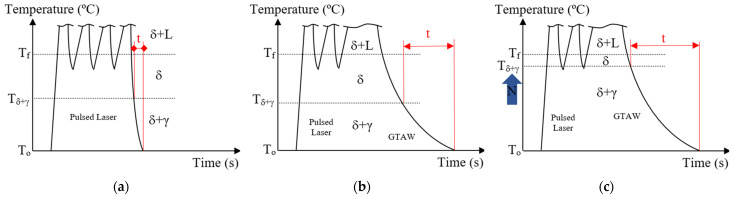
Time (t) the fusion zone remains in biphasic zone (δ + γ) while cooling in (**a**) pulsed laser Nd-YAG welding, (**b**) hybrid GTAW pulsed laser welding and (**c**) hybrid GTAW pulsed laser welding with nitrogen addition.

**Figure 3 materials-16-00543-f003:**
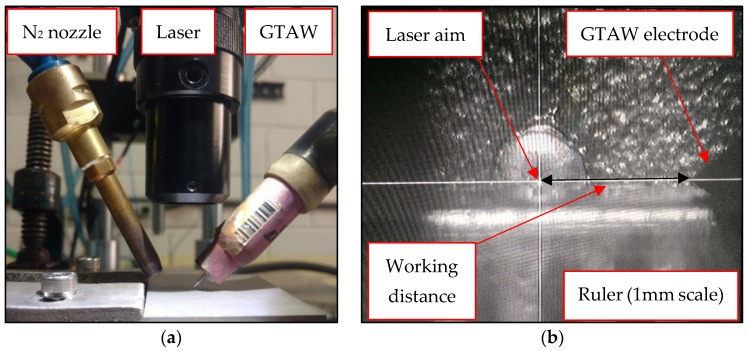
(**a**) Hybrid GTAW laser welding arrangement. (**b**) Laser camera view.

**Figure 4 materials-16-00543-f004:**
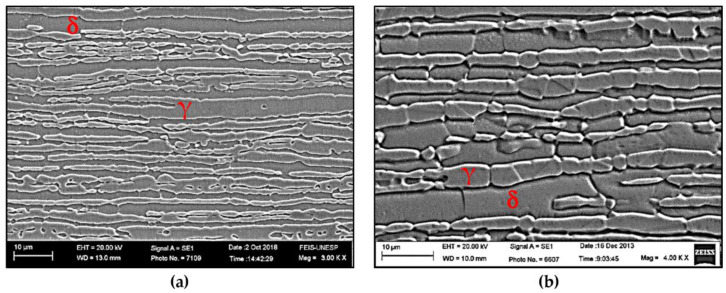
SEM pictures of the base material as received. (**a**) Reveals a balanced austenite/ferrite microstructure, with elongated geometry due to the lamination process; (**b**) Detail of austenite in high relief and ferrite in low relief due to Beraha etching.

**Figure 5 materials-16-00543-f005:**
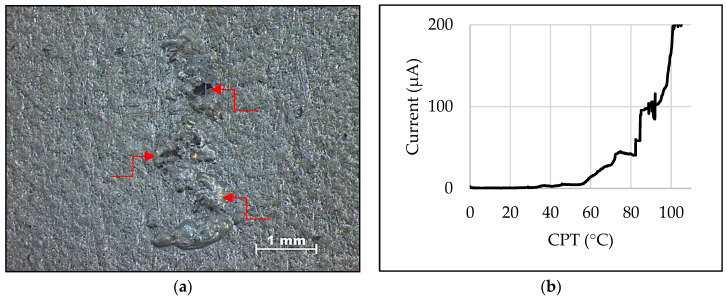
(**a**) Base metal surface after the CPT test. Arrows indicate pitting corrosion. (**b**) CPT test graph of the base metal, showing the current reaching 100 μA at a temperature of 92 °C.

**Figure 6 materials-16-00543-f006:**
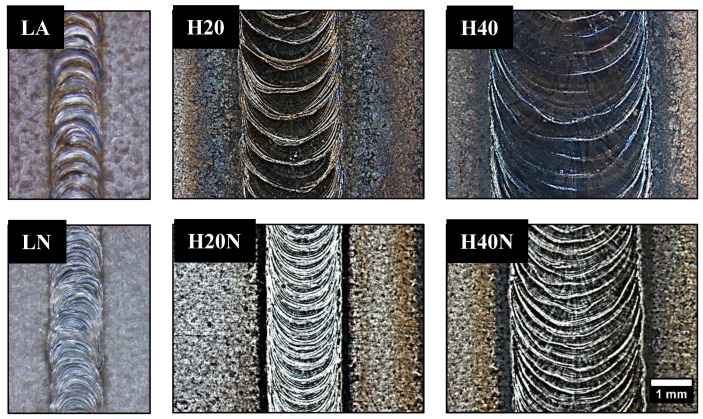
Top view of the weld beads. No visual defects found.

**Figure 7 materials-16-00543-f007:**
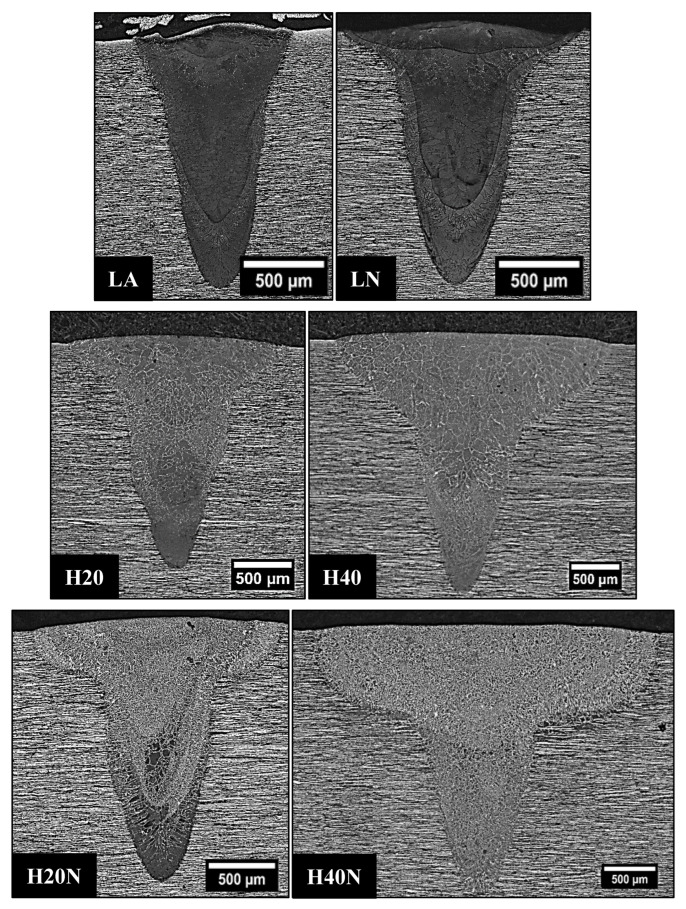
OM of weld bead cross sections. The %V of austenite in FZ increased from samples LA to H40N due to higher values of heat input and nitrogen addition.

**Figure 8 materials-16-00543-f008:**
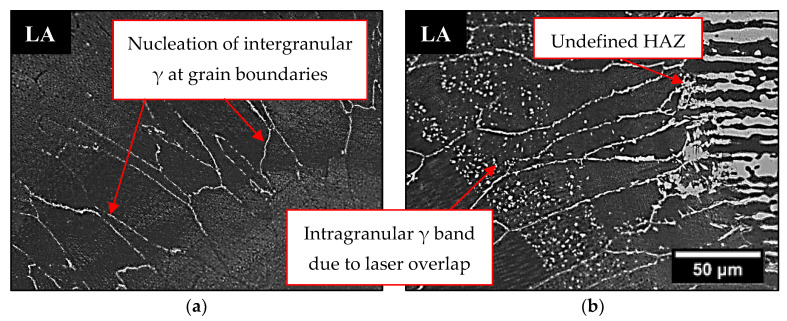
OM of LA weld bead showing (**a**) intergranular γ and (**b**) the effect of laser overlap on γ dispersion.

**Figure 9 materials-16-00543-f009:**
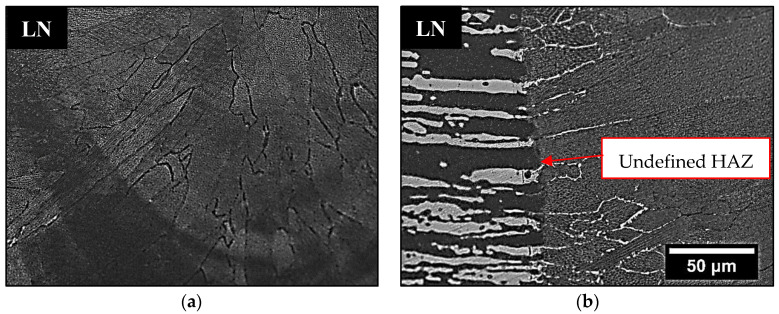
OM of LN weld bead where (**a**) the γ is imperceptible and (**b**) there is an undefined HAZ.

**Figure 10 materials-16-00543-f010:**
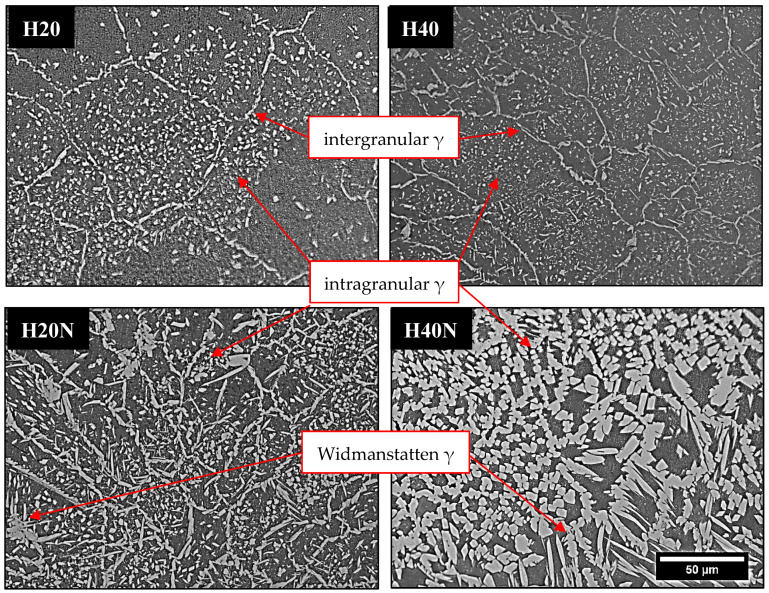
OM of weld bead sections H20, H40, H20N and H40N. The %V of austenite in FZ increased with higher values of heat input and nitrogen addition.

**Figure 11 materials-16-00543-f011:**
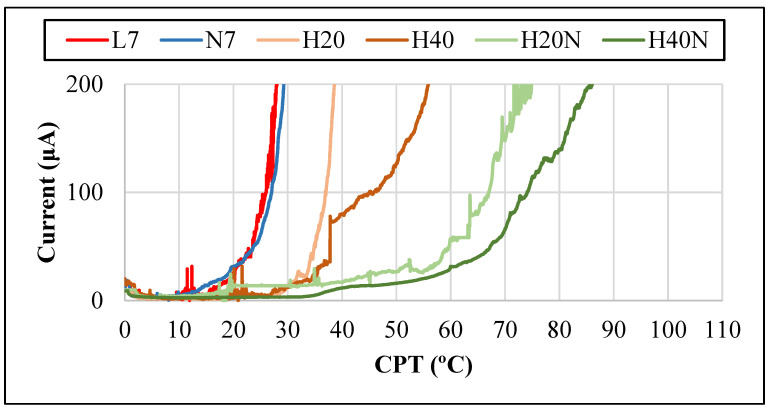
Graph with CPT test curves from each weld bead and base metal, showing the current reaching 100 μA.

**Table 1 materials-16-00543-t001:** Chemical composition of UNS S32750 super duplex steel.

Composition (%)	C	Cr	Cu	Mo	Mn	N	Ni	P	S	Si
Max	0.030	26.0	0.5	5.0	1.2	0.32	8.0	0.035	0.02	0.8
Min	-	24.0	-	3.0	-	0.24	6.0	-	-	-

**Table 2 materials-16-00543-t002:** Welding parameters.

Weld Bead	Welding Speed (mm/s)	Pulsed Laser Parameters			GTAW Parameter	Total Heat Input (J/mm)	Gas Flow Rate (L/min)	
		Frequency (Hz)	Peak Energy (kW)	Time Width (ms)	Current (A)		Ar	N_2_
LA	2	6	2.5	10	-	75	15	0
LN	2	6	2.5	10	-	75	0	18
H20	2	6	2.5	10	20	150	15	0
H40	2	6	2.5	10	40	225	15	0
H20N	2	6	2.5	10	20	150	15	18
H40N	2	6	2.5	10	40	225	15	18

**Table 3 materials-16-00543-t003:** Base metal phase balance.

Austenite (%V)	Ferrite (%V)	Standard Deviation	CPT (°C)
49.9	50.1	±0.3	92

**Table 4 materials-16-00543-t004:** Weld bead phase balance in percentage of volumetric fraction (%V); geometry obtained, with all values followed by the standard deviation (±); and CPT results.

Weld Bead	Austenite (%V)	Depth (mm)	Width (mm)	CPT (°C)
LA	7.3 ± 1.0	1.75 ± 0.03	1.38 ± 0.04	26
LN	7.5 ± 2.1	1.74 ± 0.04	1.40 ± 0.01	27
H20	11.1 ± 2.6	2.06 ± 0.06	1.86 ± 0.03	37
H40	15.8 ± 1.3	2.60 ± 0.04	2.85 ± 0.04	45
H20N	34.4 ± 12.0	2.10 ± 0.02	2.03 ± 0.02	67
H40N	51.3 ± 7.7	2.67 ± 0.02	3.54 ± 0.05	74

## Data Availability

Not applicable.

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
