# Peer review of "Increasing the Corrosion Resistance in the UNS S32750 Super Duplex Steel Welded Joints through Hybrid GTAW-Laser Welding and Nitrogen"

_materials, 2023, doi:10.3390/ma16020543_

Round 1

Reviewer 1 Report

Increasing the Corrosion Resistance in the UNS S32750 Super Duplex Steel Welded Joints through Hybrid GTAW-Laser Welding and Nitrogen discusses how to increase the corrosion resistance of UNS S32750 welds by increasing the volume percentage of austenite in the fusion zone. In general, this line of research is quite promising and interesting not only for a narrow circle of readers and specialists, but also for the general scientific community, since the problems of corrosion are very acute in materials science and metallurgy, as well as in related fields. In general, the presented study is a complete work containing a sufficient amount of experimental data and their interpretation, however, before accepting the article for publication, the authors should answer a number of questions that the reviewer had after a detailed analysis of this work.

1. The authors should provide more data on the properties of the objects they have chosen, in addition to the phase diagram of states, which generally reflects the types of compounds obtained, and also allows you to determine which concentrations should be used. Table 1 presents data on the chemical composition, however, they are given without measurement errors, and also with varying accuracy, this should be eliminated.

2. The techniques for forming welds proposed by the authors are quite interesting and promising, however, the authors should pay attention to the possibility of scaling this technology to industry, including the possibility of using other types of lasers.

3. The presented diagrams do not indicate time marks, which does not allow determining the time frame of the impact, as well as the temperatures at which these processes occur.

4. The results of corrosion tests presented in Figure 5 should be presented in a more presentable form, since the presence of pitting inclusions in the presented images cannot be determined.

5. The authors should have provided data on the elemental composition of the samples after corrosion tests.

6. The conclusion should be supplemented with further prospects for research.

Reviewer 2 Report

Review points

Article: Increasing the Corrosion Resistance in the UNS S32750 Super Duplex Steel Welded Joints through Hybrid GTAW-Laser Welding and Nitrogen.

I would like to thank you for giving me the opportunity to read this paper. There is a great deal of interest in the paper, as well as relevant discussions. Please find below some questions and suggestions that I hope will help you improve the document.

The authors presented a hybrid GTAW and laser welding under a nitrogen environment to enhance the corrosion resistance of UNS S32750 Super Duplex Steel –

1.     What is the novelty of the work?

2.     On what basis authors selected input parameters?

3.     How many sets of experiments were performed?

4.     Experiments conducted under Nitrogen environment only or without nitrogen environment also clarification required

5.     Most of the cited references are outdated and suggested to use recently published ones to describe the necessity of conducting the present work?

6.     How many experiments were conducted?

7.     On what basis experiments are planned?

8.     Citation needs to be provided for Figure 2.

9.     Suggested providing experimental setup?

10.  Provide details of measuring equipment used for analysis?

11.  Conclusions are not providing focused research findings from the work and it is suggested to rewrite

12.  English required to be improved through out the paper

Reviewer 3 Report

- What is the innovation of the authors in this research?

- In section 2, explain more about how GTAW torch, aser turret and N2 nozzle work simultaneously.

- In Table 2, how was the selection of welding conditions?

- In Figure 5-a, it should be clear what the drawn arrows refer to?

- The interpretation done in Figure 7 is not enough. The authors should provide a more complete and accurate analysis with the help of other obtained results.

Round 2

Reviewer 1 Report

The authors answered all the questions, the article can be accepted for publication.

Reviewer 2 Report

The authors have improved the article as per reviewer comments. the article can be accepted now.

Reviewer 3 Report

The manuscript can be accepted in the present form.